

# Incorporating historical information to improve extreme sea level estimates

Leigh R. MacPherson[1], Arne Arns[1], Svenja Fischer[2], Fernando J. Méndez[3], Jürgen Jensen[4]

[1]Faculty of Agricultural and Environmental Sciences, University of Rostock, Rostock, 18059, Germany
[2]Institute of Hydrological Engineering and Water Resources Management, Ruhr-University Bochum, Bochum, 44801, Germany
[3]Departamento de Ciencias y Técnicas del Agua y del Medio Ambiente, E.T.S.I de Caminos, Canales y Puertos, Universidad de Cantábria, Santander, 39005, Spain
[4]Research Institute for Water and Environment, University of Siegen, Siegen, 57076, Germany

*Correspondence to*: Leigh R. MacPherson ([leigh.macpherson@uni-rostock.de](mailto:leigh.macpherson@uni-rostock.de))

**Abstract.** Extreme value analysis seeks to assign probabilities to events which deviate significantly from the mean and is thus widely employed in disciplines dealing with natural hazards. In terms of extreme sea levels (ESLs), these probabilities help to define coastal flood risk which guides the design of coastal protection measures. While tide gauge and other systematic records are typically used to estimate ESLs, combining systematic data with historical information has been

shown to reduce uncertainties and better represent statistical outliers. This paper introduces a new method for the incorporation of historical information in extreme value analysis which outperforms other commonly used approaches. Monte-Carlo Simulations are used to evaluate a posterior distribution of historical and systematic ESLs based on the prior distribution of systematic data. This approach is applied at the German town of Travemünde, providing larger ESL estimates compared to those determined using systematic data only. We highlight a potential to underestimate ESLs at Travemünde

when historical information is disregarded, due to a period of relatively low ESL activity for the duration of the systematic record.

## 1 Introduction

Since the mid 20[th] century losses from natural hazards have been trending upwards as a result of physical and socioeconomic changes (Okuyama and Sahin, 2009). This trend is no more apparent than at the coast where concentrations of people and

assets are highest and natural hazards are more frequent and intense (Kron, 2013). The most common of these coastal hazards is flooding due to extreme sea levels (ESLs). As of 2010, up to 310 million people and assets totalling US$11 trillion were exposed to a 100-year ESL event globally (Hinkel et al., 2014). Under no adaptation and a high-emissions RCP8.5 scenario, by 2100 global population and assets at risk of flooding will increase by 52% and 46% respectively compared to the present day (Kirezci et al., 2020). Coastal flood losses by 2100 may account for 10% of global gross domestic product,

emphasizing the need for coastal risk management and adaptation (Hallegatte et al., 2013).



Risk can be defined simply as a function of probability and consequence (Lavell et al., 2012), and is thus determined in part by the likelihood and magnitude of the hazard. Further determinants of risk include the number of exposed people and assets, and their vulnerability to the specific event (Cardona et al., 2012). Reducing the impact of ESLs is done by reducing one or more of the three factors of risk: hazard, exposure and/or vulnerability (Lavell et al., 2012). While there are many approaches available for reducing coastal flood risk, each carry their own limitations and range in cost and effectiveness. Efficient coastal flood risk management aims to reduce both the investment and maintenance costs of coastal flood defences and any damages that may occur (van der Pol et al., 2021). To evaluate the feasibility of different approaches, probabilistic risk analysis may be used (Lavell et al., 2012). Therefore, estimates of ESLs must be made with a certain level of accuracy to improve flood risk analyses and ensure efficient coastal flood risk management.

Estimates of ESLs provide information on both the magnitude and probability of potentially damaging events. Such estimates may be made using extreme value analysis (EVA), a branch of statistics which seeks to assign probabilities to events that deviate significantly from the mean (Coles et al., 2001). As such, EVA is widely employed in disciplines dealing with natural hazards and probabilistic risk analysis. In regards to ESLs, uncertainties in the estimates produced by EVA are a major source of uncertainty surrounding the estimation of expected annual flood damages in the short term (before 2040; Rohmer et al., 2021) and are highly dependent on the length of data used. From a coastal management perspective, it is necessary to consider not only the likely range of estimates, but also the lower probability scenarios which lie in the upper bounds of the uncertainty range (Hinkel et al., 2015). Haigh et al. (2010) show that approximately 30 years of data are necessary to produce accurate estimates of high-end extremes (100 year return period), however this value is highly dependent on site characteristics and methods used. Arns et al. (2013) show that ESL estimates made with 30 years of data are consistent with those made using 100 years of data but highlight the affect a single large event may have on the stability of ESL estimates. For example, Dangendorf et al. (2016) find that a single large ESL event omitted from a record (in their case, 44 years) can significantly affect high-end ESL estimates. They suggest that the incorporation of historical ESL information may improve current state-of-the-art EVA, making it less prone to uncertainties arising from short water level records.

While systematic records typically extend a few decades into the past, historical records of ESLs may provide information for several centuries. Such information is not measured systematically, and is sourced via various methods (news articles, flood water marks, eye witnesses) (Jensen and Töppe, 1990). A caveat of using historical records for EVA, is that they usually only consist of very large events that would have been considered noteworthy at the time. As such, they provide important information especially for the right tail of the distribution and may reduce uncertainty in the estimation of ESLs in this range. On the other hand, extreme events which were not considered noteworthy but are still necessary for the application of traditional EVA are missing from the records. Despite this, methods exist for the incorporation of the available



historical information in the modelling of extremes (Benito et al., 2004; Prosdocimi, 2018), and several studies point to the added value these methods provide (Benito et al., 2004; Bulteau et al., 2015; Hamdi et al., 2015; Van Gelder, 1996).

In this paper we present and test a new methodology for the incorporation of historical information in EVA, which outperforms current approaches. To demonstrate this method, historical ESL information at Travemünde on the German
Baltic Sea coast are combined with systematic sea level data to obtain ESL estimates. These estimates are higher than current design standards for the region due to the incorporation of historical ESLs which are significantly larger than those present in systematic records. In fact, by incorporating historical information we show that the past 100 years of ESL activity at Travemünde has been potentially much lower than previous. As the current systematic sea level record at Travemünde is ~70 years in length, design water levels estimated using only these data may be underestimated.

**2 Background**

**2.1 Extreme Value Analysis**

Extreme events are commonly referred to by their return period, which defines an average or expected interval between exceedances of a given magnitude, typically in years (Coles et al., 2001). The annual return period $T$ of an extreme event is simply the inverse of its annual exceedance probability ($1/P$). In this study, we use standard techniques as defined by
MacPherson et al. (2019) for estimating ESLs in the German Baltic Sea region. In this section, we briefly summarize the main steps of EVA; 1) detrending, 2) sampling and 3) distribution fitting (see Arns et al. (2013) for more information).

The purpose of detrending water level data before conducting EVA is twofold; first, a fundamental assumption of extreme value theory is that the sampled extremes are stationary (Coles et al., 2001); and second, changes in water levels such as
those induced by climate change can be adjusted so that the sampled data reflects current conditions (Arns et al., 2013). Stationarity, roughly speaking, refers to a process whose statistical properties do not vary in time. As sea levels are influenced by seasonal and other long-term trends (e.g. sea level rise), stationarity is not guaranteed. While recent techniques have overcome the strict assumption of stationarity for the modelling of hydrological extremes (Calafat and Marcos, 2020; Cheng et al., 2014; Méndez et al., 2006; Menéndez et al., 2009; Mudersbach and Jensen, 2010; Serafin and Ruggiero, 2014;
Vousdoukas et al., 2016), the approach of removing non-stationarities through detrending is still widely used (Arns et al., 2015; Bernardara et al., 2011; Dangendorf et al., 2016; Haigh et al., 2014b, 2014a), especially when a deterministic attribution of the non-stationarity is not possible. However, parameter estimates of extreme distributions made using this approach reflect only the current state of extremes and do not vary in time. Uncertainties surrounding future extremes are thus typically dealt with through the inclusion of a climate surcharge (MELUR, 2012; StALU MM, 2012)






The next step in EVA is to sample the extreme events, for which there are two main approaches. The first approach known as the block maxima (BM) approach extracts a specific number of maxima within data blocks of equal length. One downside of the BM approach is that it can be wasteful, discounting extremes if multiple events lie within any one block (Arns et al., 2013). Furthermore, it is possible that the analysis is biased by the inclusion of moderate values if the block size is too small

or the dataset contains long periods of non-extremes. The more efficient peak over threshold (POT) approach, which is applied here, selects all peak events which exceed a certain threshold and provides a dataset with a smaller sampling variance (Cunnane, 1973). However, methods to determine an appropriate threshold are data dependent and can be subjective (Coles et al., 2001). Consequently, care must be taken to select an optimal threshold, as the analysis may be biased by the inclusion of dependent or non-extreme values when a threshold is set low, or through the exclusion of extreme

events when a threshold is set high (Arns et al., 2013).

Lastly, a distribution is fitted to the sampled extremes to assign probabilities to each event. This can be done empirically, for example by using the equation defined by Gringorten (1963):

$$R = \frac{i-0.44}{N+0.12} , \tag{1}$$

where $R$ is the probability of exceedance, $i$ is the rank of the event from smallest to largest and $N$ is the total number of events. This and several other approaches are described in Stedinger et al. (1993). However, a more practical approach is to employ parametric distribution functions which allow for inferences to be made on the sampled population and are not limited to return periods smaller or equal to the observation length. Within this paper we employ the Generalized Pareto distribution (GPd), which is commonly applied to the modelling of hydrological extremes and has statistical justification as

the limiting function of POT series due to the Pickands-Balkema-de Haan Theorem. Further, MacPherson et al. (2019) show that the GPd provides the best fit for ESLs at our case site of Travemünde. The probability density function of $x \sim \mathrm{GPd}(\mu, \sigma, \xi)$ is:

$$f_{(\mu,\sigma,\xi)}(x) = \frac{1}{\sigma}\left(1 + \frac{\xi(x-\mu)}{\sigma}\right)^{\left(-\frac{1}{\xi}-1\right)} , \tag{2}$$

for $x \geq \mu$ when $\xi \geq 0$ and $\mu \leq x \leq \mu$-$\sigma/\xi$ when $\xi < 0$, where $\mu$ is the location parameter, $\sigma > 0$ the scale parameter and $\xi$ the

shape parameter.

## 2.2 Increasing available sea level information

Increasing the data used in EVA is a common approach for reducing uncertainties in the estimates of extremes. In hydrology, numerical models can extend sea level information both spatially and temporally by simulating the propagation of water over some domain. Where forcing data exists (atmospheric pressure, winds, tides), hydrodynamic numerical models can produce

accurate simulations of sea levels with high spatial resolution (Arns et al., 2015; Haigh et al., 2014b), bridging the gaps in



observational sea level records. While the data provided by numerical models can be extensive, model setup is not trivial and simulations can be computationally expensive, requiring long run times. Furthermore, model simulations are dependent on the availability and quality of forcing data, which limits the period over which sea level simulations may be performed.

Another approach to extending available data, and thus reduce uncertainties in the estimation of extremes, is to incorporate historical information (Benito et al., 2004). Such approaches have also been used to account for outliers which are difficult to reconcile in the results of standard EVA (Bulteau et al., 2015; Hamdi et al., 2015; Van Gelder, 1996). The main issue regarding the incorporation of historical information with systematic data concerns the duration of observation (Prosdocimi, 2018). That is, EVA depends on a known time period in which all extremes above a threshold have occurred, and as

historical measurements are isolated data points, a duration of observation is not defined (Frau et al., 2018). Prosdocimi (2018) notes that this issue is analogous to the common statistical problem of estimating the size of a population and compares several methods available in literature, including maximum likelihood, method of moments and maximum spacing. These methods as well as graphical and Bayesian concepts were also explored by Engeland et al. (2018) when considering flooding of Norwegian catchments. Both studies discuss the added value of including historical information in

EVA, highlighting reduced uncertainties in the estimates of high-magnitude events. However, key to such analyses is the hypothesis that all historical data exceed some threshold, known as the perception threshold (Payrastre et al., 2011). While this idea seems principled, as only events greater than a certain threshold would be considered noteworthy and thus recorded, Engeland et al. (2018) raises concerns over the validity of such a hypothesis and highlights an indirect assumption that the historical data are exhaustive for the estimated coverage period.


The German Association for Water Management, Sewage and Waste (DWA) suggests three methods for the incorporation of historical extremes with systematic observations (DWA, 2012). The first method sets a perception threshold equal to the lowest historical event and considers all systematic extremes in excess of this threshold to be historical events (DVWK, 1999). A population of ESLs are created by sampling the historical extremes once and the remaining systematic events

several times equal to a weight G:

$$G = int\left(\frac{h-e'}{s-e} + 1\right),$$ (3)

where $h$ and $s$ are the lengths of the historical and systematic records in years respectively, $e'$ is the number of historical events and $e$ is the number of systematic events in excess of the threshold. The resulting population of extremes may be used to fit a parametric distribution function, from which inferences may be made.


The second method suggested by the DWA uses a Bayesian Markov-Chain Monte-Carlo (MCMC) method to maximise the likelihood of the observed sample and estimate the parameters of the distribution function (Reis and Stedinger, 2005). Samples of new parameters are either accepted or rejected based on the Metropolis-Hastings algorithm (Hastings, 1970),





where the likelihood of the new sample is compared to that of the current sample. From DWA (2012), the likelihood
function of the systematic observations and exact historical observations, given the parameters $\theta$ of a GPd, is:

$$L = \prod_{i=1}^{s} f(x_i|\theta) \cdot F(X|\theta)^{h-k} \cdot \prod_{j=1}^{k} f(y_j|\theta) \; , \tag{4}$$

where $f$ is the probability density, $F$ is the cumulative probability, $x$ is the set of all systematic observations $\{x_1, \ldots, x_s\}$, $y$ is
the set of all historical observations $\{y_1, \ldots, y_k\}$, $X$ is the perception threshold and $h$ is the length of the historical record in
years. This likelihood function can be adapted to consider a sample of non-exact historical events, with lower ($y_l$) and upper
limits ($y_u$):

$$L = \prod_{i=1}^{s} f(x_i|\theta) \cdot F(X|\theta)^{h-k} \cdot \prod_{j=1}^{k} \left( F(y_{u,j}|\theta) - F(y_{l,j}|\theta) \right) . \tag{5}$$

This allows for uncertainty to be assigned to each historical observation.

The last method suggested by DWA (2012) uses partial probability weighted moments (PPWM) to estimate the parameters
of a distribution which describes the occurrence of systematic and historical extremes. Here, the complete series of extremes
is divided into upper and lower bounded partial series around some perception threshold. Probability weighted moments ($\beta$)
are calculated by summing the PPWM of the two partial series (Wang, 1990). Estimation of the distribution parameters can
then be carried out using L-moments (Hosking and Wallis, 1997).

## 3 Data and Methods

### 3.1 Extreme sea levels at Travemünde, Germany

To demonstrate our method of incorporating historical information in EVA, we use sea level data from the German coastal
town of Travemünde. Located on the Baltic Sea coast, Travemünde has a long history of coastal flooding due to ESLs. The
first recorded ESL at the Baltic Sea coast occurred in the city of Lübeck in 1320 (Jensen and Töppe, 1990), 20 km upriver
from Travemünde. Since then, and before the introduction of systematic sea level records in 1949, a number of large
historical ESLs have occurred (Jensen and Müller-Navarra, 2008; Jensen and Töppe, 1990). Coastal defenses in Travemünde
are managed at the federal state level and have a design height equivalent to a 200 year return water level (HW200; MELUR,
2012) with an added climate surcharge of 50cm to account for future climate-induced changes such as sea level rise.
However, recent discussions between German federal states have decided that this value will be doubled to 100cm (F.
Thorenz, personal communication, June 15th, 2021). The official HW200 value is 224cm above Normalhöhennull (NHN),
the standard vertical datum in Germany (MELUR, 2012). While the current total design water level of 274cm is significantly
higher than any ESL from the current systematic tide-gauge record or in the past 100 years, this value has been exceeded in
the past, most notably during the 1872 ESL event where water levels rose to 3.4m above mean sea levels (Jensen and
Müller-Navarra, 2008). Unfortunately, MELUR (2012) do not publish their methods used to derive HW200, however from




their results it seems likely that the 1872 event is treated as an outlier and thus disregarded. Table 1 lists each historical
extreme sea level and the year in which it was recorded. For a larger sample of historical events, we consider observations
from nearby Lübeck in addition to Travemünde.

| Year | Level (m) | Location |
|---|---|---|
| 1320 | 3.10 - 3.20 | Lübeck |
| 1625 | 2.80 | Travemünde |
| 1694 | 2.65 | Travemünde |
| 1836 | 2.20 | Lübeck |
| 1867 | 1.81 | Travemünde |
| 1867 | 1.97 | Travemünde |
| 1872 | 3.40 | Travemünde |
| 1890 | 2.10 | Travemünde |
| 1893 | 1.67 | Travemünde |
| 1898 | 1.72 | Travemünde |
| 1904 | 2.22 | Travemünde |
| 1908 | 1.96 | Travemünde |
| 1913 | 2.00 | Travemünde |
| 1941 | 1.70 | Travemünde |

**Table 1. A list of historical extreme sea levels measured at Travemünde, Germany (Jensen and Müller-Navarra, 2008; Jensen and Töppe, 1990).**


For this study, we use two water level data sets from Travemünde to illustrate our method. The first is 14 historical ESL
measurements sourced from literature (Jensen and Müller-Navarra, 2008; Jensen and Töppe, 1990). Included in this dataset
are two measurements from nearby Lübeck. Second, for systematic data we use water level measurements recorded at the
Travemünde tide-gauge. The record provides 66 years of hourly sea level data, the longest available along the German Baltic
Sea coast. A comparison of extremes taken from the Travemünde tide-gauge record and historical measurements is shown in
Figure 1. The historical events are typically much larger than those recorded during the period of systematic measurements.
This suggests that either the distribution of extreme sea levels has changed over time, or the current systematic record is
insufficiently long to accurately assess high-magnitude, low-probability events. The combination of long historical and
systematic records of ESLs makes Travemünde an ideal location to test our method.





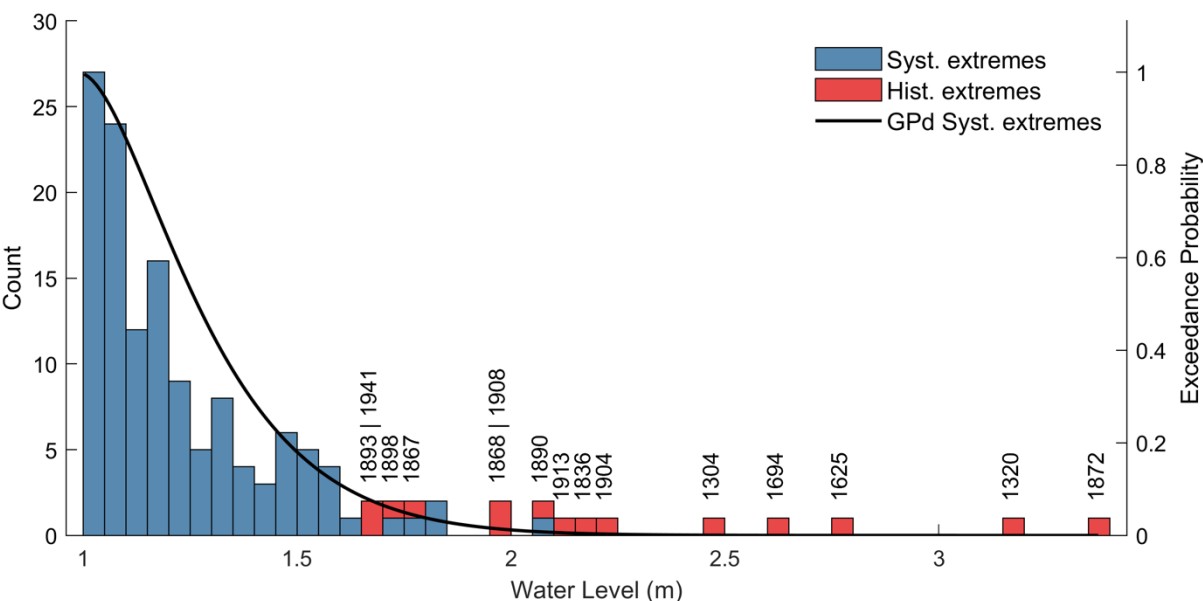

**Figure 1. Histogram of systematic and historical ESLs. Historical ESLs are shown as red bars and labelled according to their year of occurrence. Systematic ESLs are shown as blue bars with the corresponding exceedance probabilities (right y-axis) calculated using a GPd of systematic events and shown as a black line.**

### 3.2 Method

To incorporate historical information in our analysis of ESLs at Travemünde, we begin by defining a distribution of ESLs using only the systematic data, following Arns et al. (2013). First, the water level time-series is detrended using mean sea level (MSL), calculated as a 1-year moving average of sea levels. The choice of MSL allows for an easier incorporation of the historical information, which is often referenced to MSL. From the peaks of the detrended time series, extreme events are sampled using the POT technique, using a threshold of 0.98 m. This value represents the 97th percentile of high-water peaks and provides an appropriate threshold to separate extremes and non-extremes at Travemünde (MacPherson et al., 2019). A declustering period of 2 days is used to ensure each extreme is a single, independent event. Next, a generalized Pareto distribution (GPd) is fitted to the sampled ESLs using maximum likelihood estimation. Figure 2 shows the ESLs sampled from the Travemünde tide-gauge record, plotted with return periods calculated using Gringorten's equation and fitted with a GPd. Also included in Figure 2 are the historical extremes which exceed the largest ESL in the systematic record. Only these historical events are shown as their exceedance probabilities can be estimated using Gringorten's equation if we assume they are indeed the largest events for the period covering both the historical and systematic records. They are thus given a rank from 1 to 9, and the total number of historical observations is taken a 1,487, assuming ESL frequency in the historical period is the same as the systematic period (~2.1 p.a. for 710 years).





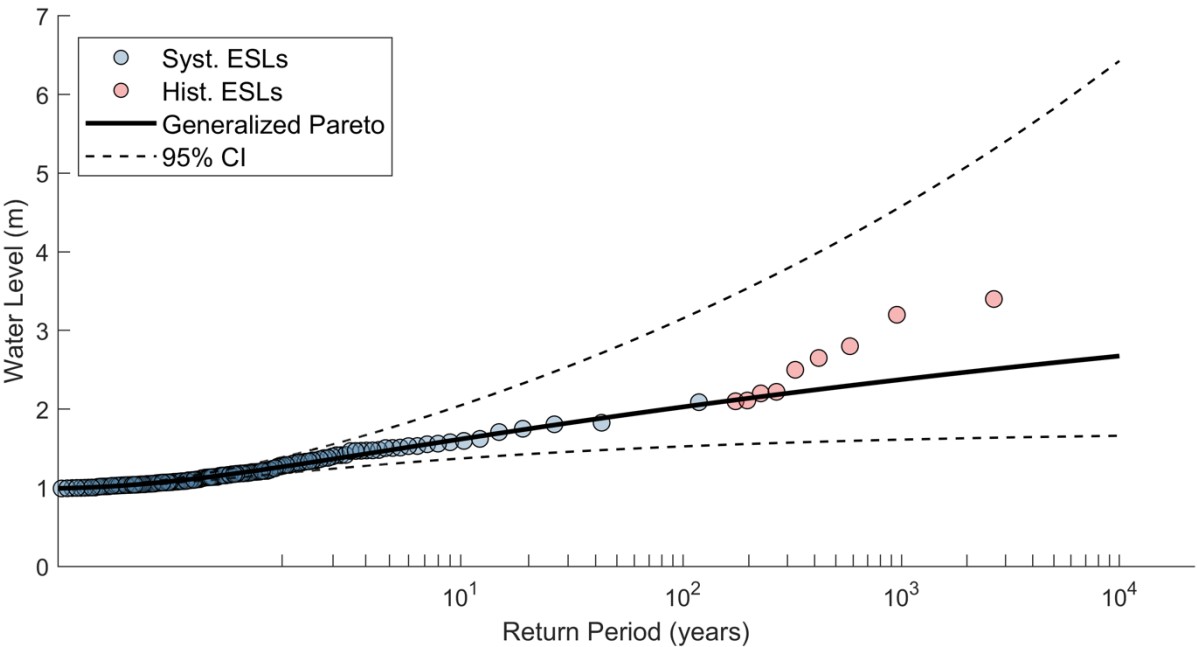

**Figure 2. GPd of systematic ESLs (blue circles) at the tide-gauge of Travemünde. Historical ESLs larger than the most extreme systematic event are shown as red circles.**

From the GPd fitted to the systematic ESLs, inferences on the underlying population of extremes may be made. However, it can be seen in Figure 2 that this distribution does not accurately describe the occurrence of historical ESLs. While it is possible that the combined systematic and historical ESLs do not form a stationary data set, another possibility is that the GPd does not fully explain the ESL environment at Travemünde. Indeed, the historical ESLs still lie well within the 95% confidence intervals of the distribution fitted using systematic data only. Hence, we assume stationarity for the period of the historical and systematic records to explore how estimates of ESLs are affected by the incorporation of historical information. The issue of stationarity is further discussed in section 4.1.

The major hurdle to incorporating historical information in EVA is the lack of a defined duration of observation. Whereas systematic data provides information on all events for a specific period, historical records define only individual events. As such, the total population of historical events is not defined and probabilities cannot be computed. To address this, we developed a method to incorporate historical information in EVA by modelling the unknown historical events. These artificial extremes are sampled stochastically from the GPd of systematic ESLs and are shown in Figure 3 along with the available systematic and historical observations. The number of artificial events generated is determined by assuming the frequency of ESLs remains constant for the period of both historical and systematic records.




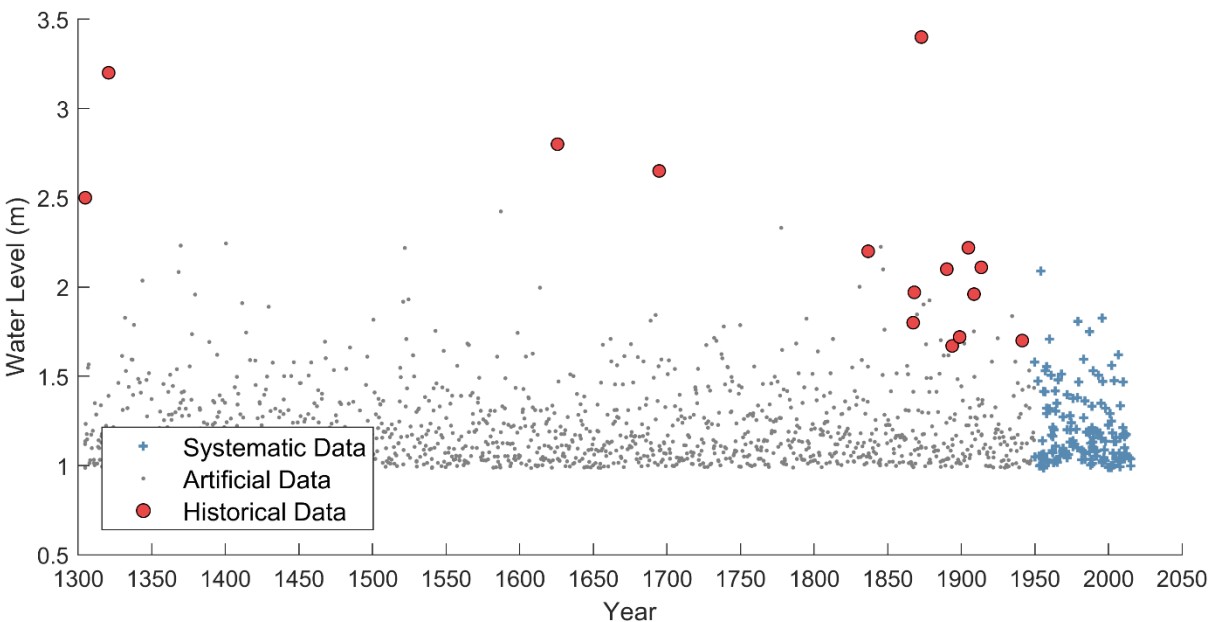

**Figure 3. All ESLs, including the observed systematic (blue crosses) and historical (red circles) data, combined with the artificial**
**events (grey dots) which were generated stochastically based on the GPd of systematic data.**

The combined systematic and artificial ESLs are all drawn from the same distribution (hereafter referred to as the initial distribution) which from Figure 2 we can see does not represent the historical ESLs well. However, as the period covered by this combined data set also covers the historical record, we can incorporate the historical information by simply substituting corresponding artificial events with known historical ESLs. An intermediate GPd fitted to this combined data set has a bias towards the systematic ESLs as the artificial events are all drawn from the initial distribution. To reduce this bias, Monte-Carlo Simulations are used to repeat the process, resampling the artificial events each time from the most current intermediate GPds.

A total of 10,000 simulations were conducted to reduce bias towards the systematic data and a final distribution is taken as the mean of all intermediate distributions. For sea levels at Travemünde, there is a large variance between high-end ESLs calculated using the intermediate distributions. However, this variance can be reduced if we further assume that no higher water levels occurred between any two consecutive observations. This assumption seems reasonable, as only events above some threshold would be considered noteworthy and thus recorded. The idea is similar to the perception threshold described in section 2.2, but its implementation is less restrictive. Whereas the perception threshold is constrained to a single value for the entire record, we assume a threshold that changes with each historical event. Therefore, we further replaced any artificial ESLs which exceed this threshold with a randomly generated value sampled from the most current intermediate distribution. Figure 4 shows a comparison of the initial and intermediate GPds from the Monte-Carlo Simulations. The final GPd has

shape and scale parameters ($\xi$ and $\sigma$) equal to the mean of those parameters from all intermediate distributions. The location

265    parameter ($\mu$) remains constant throughout.

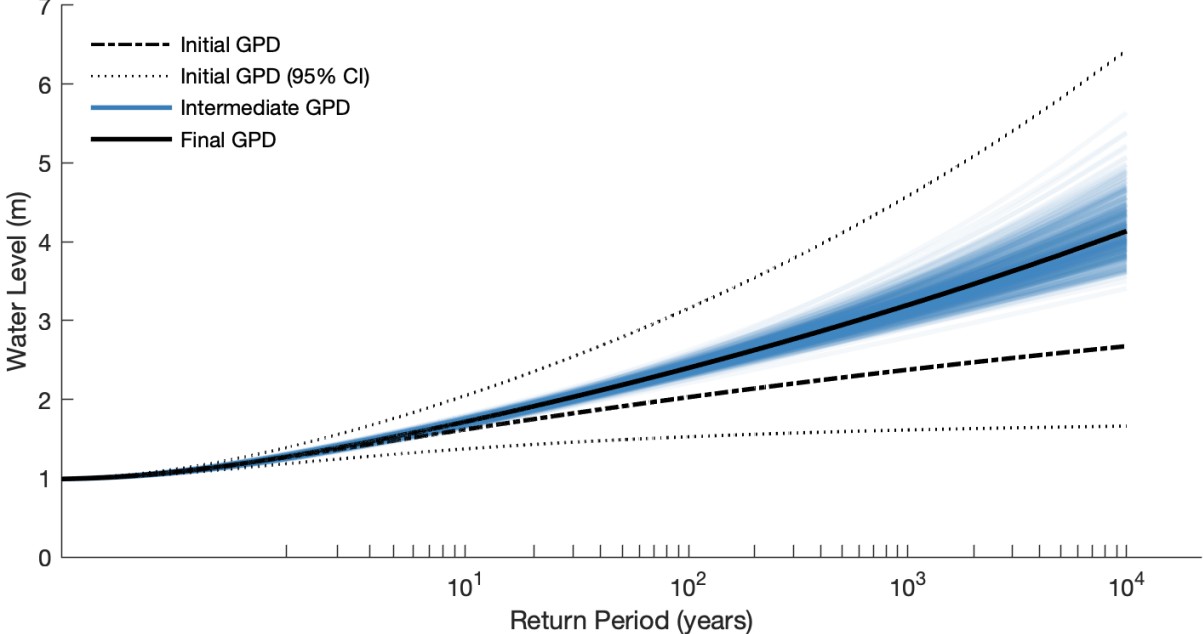

**Figure 4. GPDs of extreme sea levels at Travemünde. The initial GPd (dashed black line) was found using systematic data only, while all intermediate distributions (light blue lines) incorporate historical observations. The final GPd (solid black line) has shape and scale parameters calculated as the mean of all intermediate GPd parameters.**

270

This method does not account for uncertainties within the systematic data, as each intermediate distribution is fitted to data which contains the same set of systematic observations. To quantify these uncertainties, the analysis is performed for bootstrap samples taken from the systematic record, equal to the number of observations. We performed 1,000 iterations of the analysis using resamples of the systematic ESLs with replacement, to produce an equal number of final distributions. We

275    take the 2.5% and 97.5% quantiles of the parameters $\xi$ and $\sigma$ to define the 95% confidence intervals of the parameter estimates for our method. The location parameter ($\mu$) remains constant for all distributions. The 95% confidence intervals for both the systematic only and combined data are shown in Figure 5. For comparison purposes, confidence intervals from the systematic data only were also determined using bootstrapping.

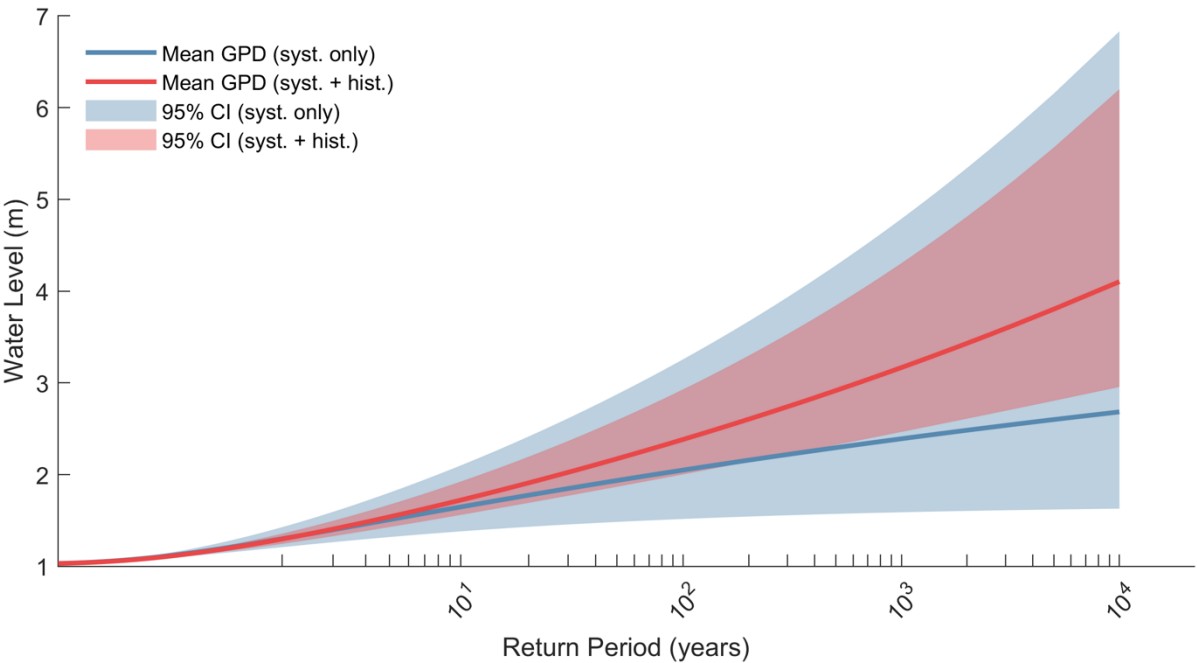

**Figure 5. Comparison of uncertainties in the estimates of ESLs using systematic data (blue) versus combined systematic and historical data (red). Bootstrapping was used to calculate the 95% confidence intervals (shaded areas) for both systematic and combined data sets.**

### 3.3 Comparison to Maximum Likelihood approach

Monte-Carlo Simulations were used to generate a large number of synthetic ESL records from which the performance of our method could be tested. We compare parameters estimated using our method with those from the maximum likelihood approach, henceforth referred to as MLA, which of the three methods suggested by DWA (2012) gave the most accurate results. Furthermore, this method is commonly used in the modelling of systematic and historical hydrological extremes (Benito et al., 2004; Bulteau et al., 2015; Engeland et al., 2018). The nature of these records was controlled by four parameters which affect the distribution of ESLs. These parameters were the shape ($\xi$) and scale ($\sigma$) parameters of a GPd, the bias between historical and systematic records ($\pi$) and the length of the historical record ($hL$). The GPd location parameter ($\mu$) was kept constant at a value of 1 m, and the length of the systematic record was taken as 100 years for all simulations. Latin Hypercube Sampling (LHS) was used to provide unique parameter combinations for 5,000 simulations and ensured that the whole parameter space was well represented. The parameters $\xi$ and $\sigma$ were sampled from the ranges [-0.5, 0.5] and (0, 0.5] respectively, while $\pi$ and $hL$ could be any integer from [1, 10] and [50, 2000] respectively. A sampling frequency of 2 events per year was assumed for both systematic and historical records, which is similar to the observed frequency at Travemünde. A complete ESL record was created for each simulation by sampling from the GPd





described by the parameters $\xi$, $\sigma$ and $\mu$, and covering both historical and systematic periods ($hL + 100$ years). From each complete record, individual ESL events were randomly sorted into systematic and historical subsets, and a bias between these was created by ensuring that the $\pi$ largest events were located within the historical record. Last, all ESLs in the historical record lower than the largest systematic ESL event were removed to emulate the missing information typical in historical data sets.

For each simulation, the historical and systematic ESL records were used to estimate the parameters of the underlying GPd using our method and MLA. Although the parameters of the underlying distribution are known, the actual distribution of sampled ESLs may differ due to the random nature of the sampling. Therefore, standard EVA was also used to estimate the parameters of the best fit distribution from the complete set of ESLs (including the removed historical ESL events). To compare the parameter estimates of each method, Bayesian Information Criterion (BIC) was used to measure goodness-of-fit. BIC is defined as:

$$BIC = k \cdot \ln(n) - 2 \cdot \ln(L) \,, \tag{6}$$

where $k$ is the number of estimated parameters (in our case, 3 for a GPd), $n$ is the total number of ESL events, and $L$ is the negative log likelihood of the estimated parameters given the complete ESL record. Figure 6 compares the results of our method to the underlying distribution and parameter estimates made using MLA for the full range of the tested parameters. As a lower BIC suggests a better fit, a negative BIC difference indicates that our method provides better estimates of the underlying distribution. The percentage of simulations where our method is preferred over MLA is also shown for intervals over the full parameter space (Figure 6. e, f, g and h).



**Figure 6. Comparison of the methods used to model the distribution of ESLs for 5,000 simulations. Boxplots show the difference in Bayesian Information Criterion (BIC) between the estimated model parameters over the tested parameters ξ (a), σ (b), π (c) and hL (d). Blue boxes group the difference between our method and the best fit distribution, while red boxes group the differences between our method and the maximum likelihood approach. A negative BIC difference indicates that our method is preferred. Histograms (Figure 6. e, f, g and h) show the percentage of simulations over the whole range of tested parameters where our method is preferred over MLA.**




Of the 5,000 simulations conducted, our method provided a better estimation of the underlying distribution than MLA in approximately 76% of cases. The performance of our method does not change significantly over the tested range of $\xi$ and $\sigma$ (Figure 6. a and b) when comparing it to the best fit distribution. However, as the bias between historical and systematic records and the length of the historical records grow (Figure 6. c and d), our method provides less reliable parameter estimates. At the same time, these estimates are increasingly improved over those provided using MLA. Interestingly, MLA

appears to slightly outperform our method when the length of the historical record is between 200 and 400 years, whereas our method shows a greater preference at all other tested values.

The improved performance of our method over MLA translates to a better estimate of HW200 in 74% of simulations. Figure 7 shows boxplots of the errors in HW200 estimates made using both methods, calculated as percentages of the HW200

values estimated using the best fit distribution. Of note is the tendency of both methods to underestimate the true HW200 value across the full range of tested parameters, with only one exception (hL < 200 years). Although MLA is capable of providing more accurate estimates of HW200 (26% of simulations), our method shows, on average, better estimates across the full range of tested parameters. Despite a slight preference towards MLA when hL is between 200 and 400 years (Figure 6. h), our method was able to provide more accurate HW200 estimates in approximately 55% of simulations. In fact, HW200

estimates for all parameter intervals determined using our method show greater accuracy compared with those made using MLA (Figure 7).

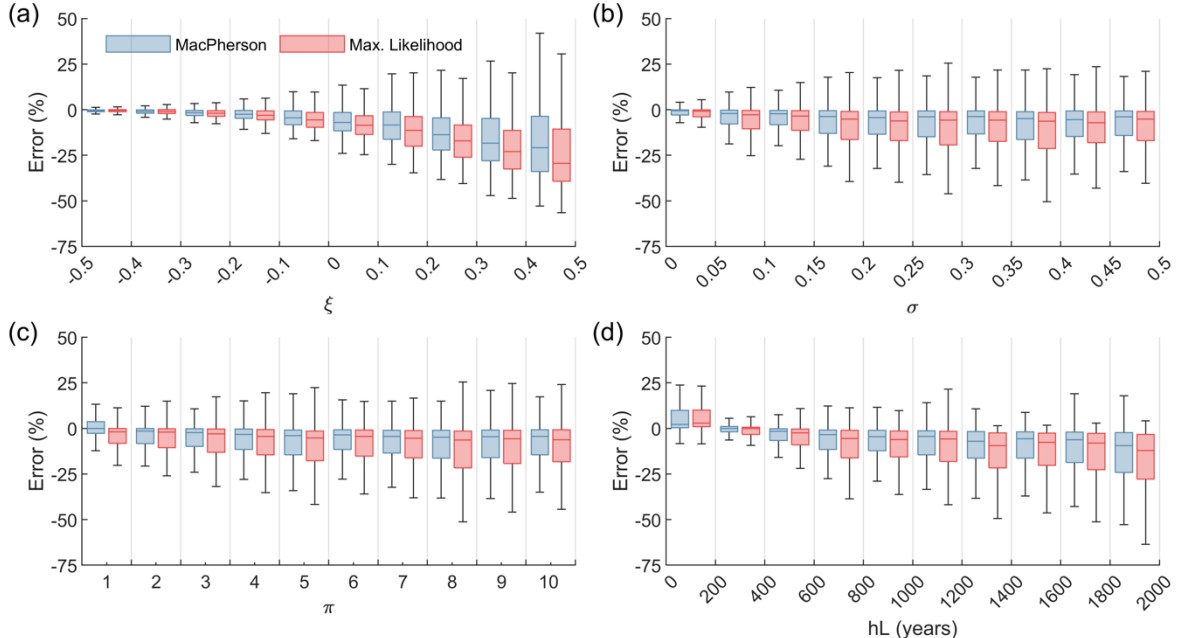





**Figure 7. Boxplots comparing the errors in HW200 estimates made using our method and MLA for tested parameters ξ (a), σ (b),**
**π (c) and hL (d). Each boxplot shows the median error, upper and lower quartiles (boxes), and the nonoutlier maximum and**
**minimum values (whiskers).**

## 4 Results and Discussion

The method outlined in this paper provides a simple approach to incorporate historical extremes into EVA, allowing for
reduced uncertainties in the estimates of ESLs and better representation of historical outliers. In addition, the method
performs well compared to the commonly used MLA (Bulteau et al., 2015; Engeland et al., 2018; Gaume et al., 2010; Reis
and Stedinger, 2005), providing more accurate estimates of ESLs over a range of tested parameters for the GPd.

### 4.1 Significance

From our case study at Travemünde, the estimates of high-end ESL have been drastically changed by incorporating historical
information. The main question that arises is whether these results better reflect the current ESL environment at
Travemünde? From Arns et al. (2013) and Haigh et al. (2010) it would seem plausible that the 66 years of systematic sea
level records at Travemünde allow for accurate estimates of high-end ESLs (~200 years). However, we know that sea levels
at Travemünde far in excess of the current HW200 estimate are possible. Furthermore, exceptionally large events have
resulted in drastic changes to the estimates of high-end ESLs in the past (Dangendorf et al., 2016). Thus, the question arises
whether the historical and systematic data sets can be reconciled, or has there been some change in the generation
mechanisms of ESLs at Travemünde which renders the historical records no longer representative?


A fundamental assumption of EVA is that the population of extremes is stationary, and while methods to model non-
stationary extremes exist, they are not yet capable of incorporating historical information as the duration of observation is
not defined. Our method and those outlined in this paper all assume stationarity between both historical and systematic
datasets. Consequently, the accuracy of ESL estimates depends on whether this assumption is true. Unfortunately, it is not
possible to determine whether both data sets are drawn from the same distribution, for the same reason we cannot simply
combine the two data sets when conducting EVA. However, a more relevant question may be: what information is needed to
satisfy coastal flood risk management? As discussed by Hinkel et al. (2015), high-end estimates rather than the those in the
likely range are of greater use when dealing with coastal flood risk from the perspective of coastal managers. Moreover,
designing coastal defences at heights significantly lower than ESLs experienced previously seems counterintuitive. From
systematic data, we estimate that the 1872 ESL at Travemünde has a return period of approximately 45 million years.
However, this extraordinary event is better represented once historical information is considered, reducing the estimated
return period to a more sensible 2,500 years. This is comparable to the case of the Mulde River in Germany, where the return
period of a flooding event in 2002 was reduced from 5.5 million years to approximately 1,000 years once historical
information was considered (DWA 2012).






While the better representation of large historical extreme events such as the 1872 ESL event is beneficial, changes to the estimates of design water levels carry greater importance. The official HW200 estimate at Travemünde is 224cm (MELUR, 2012), slightly higher than our estimate of 219cm using systematic data only. However, by incorporating historical ESLs such as the 1872 event, our value increases to 262cm. Not only is the additional 43cm a large portion of the current 100-year

climate-surcharge (50cm) used for design purposes along the German coasts (MELUR, 2012; StALU MM, 2012), but also nearly half of the future climate-surcharge value of 100cm (F. Thorenz, personal communication, June 15th, 2021). Furthermore, we showed a tendency of our method and MLA to underestimate the HW200 value when a bias exists between historical and systematic records, which appears to be the case at Travemünde. Consequently, the effective lifetime of coastal defences in the German Baltic Sea region may be severely reduced due to an underestimation of HW200.


An added benefit of incorporating historical information is the reduction in uncertainties surrounding ESL estimates. According to Rohmer et al. (2021), the reduction of uncertainties in the parameterisation of ESL distributions is one of two key areas to address in order to reduce uncertainties in the expected damages of coastal flooding in the near future (before 2040). Using only systematic data, the 95% confidence intervals range from 154cm to 367cm for the HW200 estimate at

Travemünde, a difference of 213cm. Incorporating historical information reduces this range by almost half to 116cm with a lower and upper bound of 214cm and 330cm respectively.

Another significant implication of the results of this study relates to the application of traditional EVA. A general rule of thumb in regard to EVA is that at least 30 years of data is required to estimate a 100 year event while maintaining sensible

uncertainties (Arns et al., 2013; Haigh et al., 2010). Despite the systematic record at Travemünde containing almost 70 years of data, there is a large change to the distribution of ESLs after the incorporation of historical information. A similar result was seen by Dangendorf et al. (2016) at the German North Sea coast when they included an exceptionally large event within a systematic record of ~44 years. These results highlight the uncertainties intrinsic in EVA. For the case of the German Baltic Sea, where ESLs may be generated by very specific series of physical phenomena acting over the entire Baltic Sea

(Jensen and Müller-Navarra, 2008), potentially much longer records are required to fully represent the ESL environment. Unfortunately, systematic records in the region are typically much shorter than Travemünde, and it may be necessary to extend the available information using approaches mentioned in Section 2.2, such as incorporating historical measurements.

## 4.2 Implications for coastal management at Travemünde

When comparing systematic and historical ESL measurements at Travemünde, the current systematic records appear to be

taken during a period of relatively low ESL activity. For example, the maximum recorded water level between 1915 and 2015 was 209cm in 1954. In contrast, no less than 4 events exceeding this height occurred during the 100-years prior, including the exceptionally large 1872 event which reached 3.4m. Using the Poisson distribution, we are able to determine





the likelihood of such a period of low ESL activity based on the annual exceedance probability ($\lambda$) of the largest ESL event, and the period of interest (t):

$$P(\lambda, t) = e^{-\lambda \cdot t}, \ T = {1}/{\lambda}, \tag{7}$$

where $T$ is the return period in years. We estimate $\lambda$ for an ESL of 209cm using the GPd fitted to systematic extremes and the GPd derived using the method described herein. These values are then substituted into the equation above to determine the likelihood of a 100-year period in which no events in excess of 209cm occur. When considering only systematic data, such a period of low extremes has a likelihood of 50%. However, this value drops significantly to 7% when the annual exceedance probability is estimated using our method of incorporating historical information.

At Travemünde, the HW200 value is given by MELUR (2012) to be 224cm. Over a 100-year period, the likelihood that no events exceeding this level would occur is approximately 61%. In contrast, we estimate the return period of a 224cm ESL to be approximately 65 years when historical information is also accounted for. Thus, the likelihood for an ESL to occur within a 100-year period that is in excess of the HW200 level is reduced to ~21%. In other words, the probability that the HW200 level is exceeded within 100 years is approximately doubled from ~40% to ~80%. This has clear implications for the planning and management of coastal defenses at Travemünde which are designed based on flood risk. As risk is defined as a function of probability and consequence, a two-fold increase to the likelihood of coastal flooding would dramatically influence the determination of risk, and by extension any planned coastal protection measures.

It is necessary to reiterate that the results of this method depend upon the assumption that no changes in the ESL environment at Travemünde have occurred. On the other hand, disregarding historical information in EVA requires the opposite assumption, that current ESLs follow a different distribution than those in the past. As discussed in section 4.1, perhaps the more useful approach from a coastal flood risk management perspective is the method which provides the more extreme cases, and furthermore, the method which incorporates all available data.

### 4.3 Outlook

The method outlined in this paper relies on traditional EVA, with the assumption that the data under consideration is stationary. Recent studies have overcome this strict assumption of EVA to model the occurrence of non-stationary extremes (Calafat and Marcos, 2020; Cheng et al., 2014; Méndez et al., 2006; Menéndez et al., 2009; Mudersbach and Jensen, 2010; Serafin and Ruggiero, 2014; Vousdoukas et al., 2016). Theoretically, non-stationarity could be incorporated into our method by fitting non-stationary distributions in place of the stationary techniques we employed. While this was considered outside the scope of our research and may produce large accuracy issues, it offers a first step towards future studies where historical information is incorporated in non-stationary EVA.



## 5 Conclusions

Estimates of ESLs can be improved with reduced uncertainties and a better representation of historical outliers by incorporating historical information into EVA. We present a new approach to incorporate historical data which compares favourably to other commonly applied methods when considering hydrological extremes. Whereas other methods assume a duration of observation based on a perception threshold, we simply extend the systematic data through inference of the systematic ESL distribution and substitute values where historical information is available. Monte-Carlo Simulations are

used to estimate the median parameter values of the distribution describing the combined data. A major benefit of this approach is that best practices used for the estimation of ESLs from systematic data can be maintained for the combined historical and systematic data set, which makes the application easy for practitioners, too.

Our approach assumes that the distribution of extremes is unchanged for the period of both historical and systematic records.

Indeed, this stationarity assumption is required for EVA and the estimation of ESLs. However, it is not possible to confirm whether this is true, for the same reasons EVA cannot simply be conducted using discrete data points with an unknown duration of observation. Regardless, the use of historical information without the confirmation of stationarity still has its merit in coastal risk management, given the high-risk perspective. The assumption that historical information is no longer relevant due to non-stationarity and statistical outliers may have greater consequences for coastal planning than assuming

stationarity between the two data sources. Hence, the method described in this paper provides an effective tool to reconcile historical information and systematic data.

The method was applied at Travemünde, located at the German Baltic Sea coast, where long records of systematic and historical data are available. We find that incorporating historical information results in large increases to the estimates of

design heights for coastal defences in the region. While official design heights are determined using a return period of 200 years (MELUR, 2012), we estimate the return period of current design heights to be much reduced at approximately 65 years. Furthermore, systematic measurements at Travemünde appear to be recorded during a period of relatively low ESL activity, potentially biasing analyses which include only systematic data. This highlights the importance of considering historical measurements when conducting EVA.

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
