# Peer review of "Incorporating historical information to improve extreme sea level estimates"

_Natural Hazards and Earth System Sciences, 2021_

## Referee Comment (RC1)

**Review summary:**

The article presents a new method to deal with historical information in classical extreme value analysis (EVA) of sea levels. The authors show that their method outperforms the Bayesian Markov Chain Monte Carlo (MCMC) approach, used in several papers to tackle the issue of partial historical information in EVA, in terms of estimating extreme sea levels (ESLs) at Travemünde, Germany. In addition, authors show that the estimation of 200-year return water level (HW200, from which is based the design height for coastal defenses at Travemünde) is larger with their method than the current official value, determined using only tide gauge records (aka systematic data), highlighting a possible underestimation of coastal flooding risk at Travemünde.

Overall, I think this is a good paper. It is well written and the objective, method and results are presented clearly. However, the article would benefit from some complementary information to better assess the novelty and relevance of the results.

I detail my review below, separating major from minor comments.

**Major comments:**

2.1 Extreme value analysis

- L118-120, equation(2): strictly speaking, the Generalized Pareto distribution is a 2-parameter distribution. The location parameter μ described by the authors is in fact a threshold fixed by the user. When it comes to fitting the GPd to some data, only 2 parameters are estimated in the process. In contrast, the Generalized Extreme Value distribution (GEV) has 3 parameters. There may be a confusion here, I suggest to make it clearer. Furthermore, the support of the distribution is x > μ (strictly) when ξ > 0 (strictly) and μ < (strictly) x <= μ-σ/ξ when ξ < 0. ξ = 0 corresponds to the particular case of exponential distribution.

2.2 Increasing available sea level information

- L146-173: Three methods for the incorporation of historical extremes with systematic observations are presented. I regret that the main reference here is in German (DWA, 2012) thus I was not able to check and better understand methods 1 and 3. Actually, as the only method further used in the article is method 2, I suggest not mentioning the other 2. If authors still want to keep this description, then I strongly recommend to test and compare these additional 2 methods in the study.

  Equation(4): I think there is a mistake here as the presented formula is based on a GEV distribution and not a GPd. The likelihood formulation for a GPd is more complex, see for example Bulteau et al. 2015 https://doi.org/10.5194/nhess-15-1135-2015. In a GPd framework we deal with peak event probabilities, while in a GEV distribution framework we deal with annual exceedance probabilities (in case of annual maxima). Authors should clarify this point which could lead to strong impacts on results.

3.1 Extreme sea levels at Travemünde, Germany

- L184: Authors write "The official HW200 value is 224cm above NHN". I have a concern about datum references in the text. I believe that all values in the remaining article are given in meters (or centimeters) above mean sea level (MSL) - see section 3.2 Methods. If the authors' statement is correct, I suggest converting the HW200 value in cm above MSL so that we can

compare it more easily with values in Table 1 for instance. However, it seems that the value 224cm is used in section 4.2 (see for instance L416) working with the GPd whose parameters are estimated based on systematic and historical data expressed as values above MSL. So I am a bit confused: is the statement at L184 wrong or is there a mixing of values expressed with respect to different datum references?

- L190-191 and L197-198 : Authors indicate that two measurements from nearby Lübeck are included in the historical dataset in order to make it larger. This is not discussed in the discussion section whereas that could generate more uncertainty in ESL estimates. Have authors tried to perform the analysis without these two measurements? I suspect that the first one (3.10-3.20m above MSL observed in 1320) has a huge impact on results. A sensitivity analysis could be performed. At least, this point should be discussed. Also, is there an argument or a study that would give credit to this merging of historical observations from 2 different sites into a single historical dataset? Are sea levels comparable between the two sites?

**3.2 Method**

- L255-262: I do not understand clearly this paragraph. I think another figure or table showing the problem of large variance between high-end ESLs as mentioned by the authors would help. Moreover, the authors write "this variance can be reduced if we further assume that no higher water levels occurred between any two consecutive observations". Do they mean "between two any consecutive *historical* observations"? If so, this should be clarified.

  In the process, authors replaced "any artificial ESLs which exceed [the moving threshold] with a randomly generated value sampled from the most current intermediate distribution": what if the new value still exceeds the threshold? Clarification is also needed about the moving threshold: for example let's say we have ESL1 larger than ESL2. Between these two consecutive events, is the perception threshold equal to ESL1 or ESL2?

  Authors indicate that in the classic Bayesian MCMC-MLA approach, a perception threshold is constrained to a single value for the entire historical record. This is true but it is also known that what matters most in an EVA combining systematic and historical data is not the number of historical events but the length of the historical period and the fact that we have an exhaustive dataset above the perception threshold (see for instance Payrastre et al., 2011 https://doi.org/10.1029/2010WR009812). Actually, it might lead to better results to set a higher perception threshold and ensure exhaustivity even if the historical events dataset must be reduced accordingly. Sensitivity tests would be interesting to compare authors' method with MLA with a perception threshold equal to 2.5m/MSL (1304 event) or higher. That way, the exhaustivity requirement for MLA would be better fulfilled and the comparison would be fairer.

**3.3 Comparison to Maximum Likelihood approach**

- Whereas the beginning of that section belongs well to global section 3.Data and Methods, I suggest to move the part from L312 to L345 to a new Results section.
- L311: in Equation(6), k is the number of estimated parameters. Once again, there are only 2 parameters in a GPd, not 3.
- It is not clear to me what is the value of the perception threshold used in the MLA. Clarification is needed. As mentioned above, tests with different perception thresholds would be interesting to conduct.

**4 Results and discussion**

- This global section should be renamed "Discussion" and all content related to the results should go to a new Results section. The Results section should contain: 1) results of comparison between MLA and the proposed method, 2) estimations of HW200 using the proposed method and comparison with current official values.
- L361-363: This statement about non-stationarity in EVA incorporating historical data seems to be in disagreement with section 4.3 Outlook. From the one hand authors say methods are not yet capable of mixing non-stationarity and historical information, and on the other hand, they claim that theoretically there is no obstacle. Clarification is needed.
- It is important to highlight that the proposed method cannot apply to uncertain data. Only historical data with known values can be dealt with as the estimation of GPd parameters is performed the classical way (besides it is put forward by the authors in the conclusion as an advantage). However, historical information is often partial and uncertain. In many other places in the world, one can only access to lower bounds of historical ESLs or ranges of values (see for example the 1320 event in Table 1), and exact values are rare. Only a Bayesian framework is able to deal with this issue and to properly incorporate uncertainty in the EVA. This could be a limitation of the study and the proposed method as it may not apply everywhere. This point should be discussed.

**Minor comments:**

- I suggest the title of the article to be changed. It does not reflect the novelty claimed by the authors in the text. Moreover, it is not shown nor proved in the article that ESLs estimates are *improved* while incorporating historical information: authors themselves say that their method lead to *larger* estimates than current official values underlining the crucial assumption of stationarity to end up with this result, assumption which cannot be confirmed. In a risk prevention perspective, that might be laudable, but it cannot be said that ESL estimates are improved as we do not know that for sure.
- L15: Authors write "This paper introduces a new method for the incorporation of historical information in extreme value analysis which outperforms other commonly used approache**s**." In fact, only the Bayesian MCMC - or MLA approach is compared with the authors' method. This sentence should be revised accordingly.
- L17: Authors use terms such as "posterior" and "prior" distributions. These are commonly used under a Bayesian probabilistic framework. As this is not the case here, I suggest rephrasing to avoid confusion.
- L50: dash is missing (100-year return period)
- L110: there is a mistake in the definition of i : i is the rank of the event ranging from largest to smallest, or in other words in descending order, not the opposite.
- L115: "(…) due to the Pickands-Balkema-de Haan Theorem." A reference should be added here.
- L121-129: Authors discuss the possibility of increasing available data to perform an EVA by using sea levels outputs of numerical models in addition to tide gauge data. They expose some issues such as long run times and forcing factors that must be sound to get relevant results. In addition, authors could mention the issue of combining heterogeneous data in EVA: tide gauge data and model outputs do not carry the same uncertainty, which can lead to errors hard to quantify in ESL estimates. One possibility to deal with this, is to explicitly incorporate uncertainty on modeled values in the EVA through a Bayesian framework (see for example Nicolae Lerma et al., 2018 https://doi.org/10.5194/nhess-18-207-2018 ).

- Table 1: column 2 should be relabeled as follows "Level (m above MSL)". The legend should be modified as follows: "A list of historical extreme sea levels measured at Travemünde and Lübeck, Germany) (refs)." Also there are two events occurring in 1867, and this is not in accordance with Figure 1 (one event in 1867, one event in 1868).
- Figure1 & Figure3 & Table1: There is a 15th historical event in Figures 1 and 3 (occurring in 1304) which is absent from Table1.
- L213: Authors may add here that all water levels expressed in the following are expressed in meters with respect to MSL. That would clarify and avoid confusion when datum reference is not mentioned.
- L216: "generalized" should be written with a capital letter "Generalized"
- L222: word should be replaced by the one in bold "(…) the total number of historical observations is taken **as** 1,487, assuming(…)"
- L223: a word is missing "(…) is the same as **in** the systematic period (…)"
- Figure2: y-axis should be relabeled "Water Level (m above MSL)"
- Figure3: how dates are attributed to artificial events? Is the Poisson process modeled in the stochastic generation?
- L291: Here is the first time authors mention the bias between historical and systematic records. Definition should be given here.
- Figure 6: explanation of boxplots is missing in the legend (cf. legend of Figure 7).
- L337: Authors claim that MLA is capable of providing more accurate estimates of HW200 in 26% of simulations, or that their method provides better estimates in 74% of simulations (L332). I do not see where those figures come from.
- Figure 7: Adding a horizontal line at 0% of error on each graph would make it easier to read.
- L367: word in bold must be deleted "(…) high-end estimates rather than **the** those in the (…)"
- L381: a personal communication is used as a reference for the future climate-surcharge value of 100cm. Does this refer to a design constraint specific to Travemünde or German coasts? If so, when will this new value be implemented?
- L406: rephrasing suggestion: "In contrast, no less than 4 events exceeding this height occurred during the **previous 100 years.**"
- L408: rephrasing suggestion: "(…) of the largest ESL event **in the systematic period**".
- L417: Authors write "Over a 100-year period, the likelihood that no events exceeding this level would occur is approximately 61%". Is this result obtained using systematic data only ?
- L419: The sentence is not clear. I suggest rephrasing: "Thus, the likelihood that no event exceeding this level would occur within 100 years is reduced to ~21%"
- L423: "a two-fold increase **in** the likelihood"

References:

- Numerous journal names are missing in references (for examples: L475, L477, L482, L489, L500, L504, L510, L513, L515, L518, L523…)
- Sometimes journal name is written entirely sometimes only with abbreviations. Authors must follow nhess standards.

---

## Referee Comment (RC2)

**Review report for "Incorporating historical information to improve extreme sea level estimates"**

**1   Summary**

The article by MacPherson et al. presents a method to incorporate historical Extreme Water Level (ESL) information into classical Extreme Value Analysis (EVA) using a Bootstrapping approach. They demonstrated the method for Travemünde, Germany and indicated to a possible underestimation of the current design water level for the flood defense.

The analysis of this paper is generally well-written. However, I find that the discussion is very site-specific, and many potential avenues for application of such a method in other places are not touched.

I recommend acceptance of this article after clearing-up some of the details - particularly regarding the historical data - which is essentially the main attraction here. My comments are given below. LXX means the corresponding line number.

**2   Major comments**

**2.1   On the study area**

- A study area map is needed for the ease of the readers.

- I think Section 3.1 would be a easier read if the datums are first identified. For example, please consider indicating NHN in-terms of mean sea level (MSL) and the corresponding value of HW200, HW200+50cm etc in terms of MSL. Is NHN is equal to mean sea level?

- I think it is worth mentioning in the text that the tide condition, e.g., the fact that the low tidal range at Travemunde allows directly looking into the ESLs. I think, it would not be so straightforward without this micro-tidal setting. L203.

**2.2   Historical ESL events**

- L130-145 discusses the approach of incorporating the historical ESL, and gives a background. The "main issue" of incorporating such data is identified to be the fact that historical measurements are isolated data points, and not having the duration of the observation defined. While these are true, I think one other major element is not discussed here, or other places - is the consistency of the data itself (in terms of datum). When stretched backwards, even with systematic data - such as tide gauge - datum consistency can get quite tricky. As you have already discussed before, presence of a large event can have significant impact on the EVA, I believe this point needs further attention in the article. Essentially, I believe this might have been discussed in the chapter (?) by Jensen et al.

- Table 1. Please consider a indicating for each storm where they are sourced from.

- Which value of 1320 is taken? 3.10 or 3.20? As it is a large value, what do you expect in terms of uncertainty? More important question is related to point 1 - how comparable is this measurement in-terms of datum/height accuracy?

- L190: From a quick look at Jensen et al., for 1694 event there is a 11cm difference in ESL between Lubeck, and Travemunde. For the events, where Lubeck value is taken for the sake of extending the series, could it induce another set of bias? To put it differently, are the sea levels comparable between the two sites?

- I did not find the 1304 event in the table, which is shown in Figure 1. In Jensen et al. this event exists but no date nor height is reported.

- Is it necessary to consider the VLM corrections in the historical series? Particularly given that we are taking values from 700 years ago. As in L211, a detrending is done for the systematic data. It would definitely create a question regarding if you need to apply some corrections to the historical series too.

**2.3 Comparison with maximum likelihood approach**

- Among the 3 methods of DWA (2012), only method 2 seems to be compared to. If you do not consider the other methods to be tested, please remove them from the description to make it lean. Without multiple comparison, L15 needs a revision - consider changing from ". . . outperforms other commonly used approaches." to ". . . outperforms currently used approach for Travemunde."

**3 Minor comments**

- L18-19: please consider adding how much larger ESL estimates (in percentage?)

- L28: "will increase" → "projected to increse"

- L94: Could you please provide a more accessible literature on the inclusion of "Climate Surcharge", instead of these two German refs (reports, I presume)?

- L115: missing ref to Pickands-Balkema-de Hann theorem.

- L151: what does "int" mean?

- L296: what is the value of observed frequency?

- Please be consistent on the definition of the level (m above MSL) throughout the MS.

- L404: could this be cross-referenced to previous studies? (e.g., Catelog by Jensen et al.).

- L414: please consider replacing "significantly" → "substantially", as there is an expectation to see a significance test when the term significant is used in the context of probabilistic analysis.

- L416: . . . 224 cm. . . please see point on the consistent identification of the datum.

- L424: are the floodable regions is under insurance-schemes? Could be a good addition their interest on such results.

---

## Referee Comment (RC4)

**Overview:**

I think this paper addresses an important issue for statistical modelling of extreme values and the characterization of coastal flooding hazard, which is to partial overcome the inevitable scarcity of data on extreme events and to account for outliers by incorporating historical information. Indeed, this paper shows how the use of historical information can improve the probabilistic and statistical modeling of extreme sea levels. This has been achieved by combining systematic (observed & artificial) ESLs with historical information using a bias reducing method based on a MC resampling method. I can see that this work could be used in many applications and bring improvement on the approaches already available for such applications.

On the whole, I felt able to follow the proposed method. I did, however, reach the end of subsection 3.2 and realize that I was uncertain about the used method of bias reduction (from the initial intermediate GPd to the last one) and how the first intermediate GPd can be used to continue the processes of bias reduction.

**General comments:**

While a potentially useful method that could reduce uncertainties and in-crease the reliability of extremal estimates, there are, however, at least four major comments to which authors have to provide real and concrete answers, primarily to ensure that the usefulness of the authors' proposed method is conveyed at a standard that is consistent with that of the underlying concept:

1- I wonder if the proposed method provides a real improvement relative to the classic approach (the use of a threshold of perception method and the maximum of likelihood applied to both systematic and historical data, especially if we know that gathered hist information is exhaustive and represents the largest values during the historical period.

2- Regarding the first intermediate GPd, authors had to incorporate the historical information by developing a truncated GPd conditional to the largest historical data instead of simply substituting corresponding artificial events with known historical ESLs. Authors can refer to the censoring approaches proposed by Parent and Bernier (2003) (Parent, E., Bernier, J., 2003. Bayesian POT modeling for historical data. J. Hydrol. 274 (1–4), 95–108.)

3- The first "intermediate" GPd is not really intermediate. According to me, it is a distribution completely different from the initial one. A different object and cannot be used to continue the process of bias reduction used in this paper.

4- Several methods of bias reduction are presented in the literature. Authors are invited to give more details about the method of bias reduction used in this paper and give a quantitative evaluation of this bias.

**Specific comments:**

**The issue regarding the duration of historical period:**

Lines 135-138: "Prosdocimi (2018) notes that this issue is analogous to the common statistical problem of estimating the size of a population and compares several methods available in literature, including maximum likelihood, method of moments and maximum spacing". The relationship between the maximum likelihood, method of moments and maximum spacing and the main issue regarding the duration of observation related to the incorporation of historical information with systematic data, is not clear for me. In addition, the authors sated in the following sentence (lines 138-139) that the same issue was considered by Engeland et al. (2018): "these methods as well as graphical and Bayesian concepts were also explored by Engeland et al. (2018) when considering flooding of Norwegian catchments". This is not true because for some watersheds, Engeland et al. (2018) followed Prosdocimi (2017) and set the length of the historical period to be the time span from the fitst historical event to the end of the historical period plus the average time spacing between the historical events and another subjective method for the rest of watersheds.

**Systematic observations exceeding the threshold of perception**

Lines 145-155 The authors stated this idea (including all the systematic observations higher than the threshold of perception in the historical information) was suggested in 2012 by the German Association for Water Management, Sewage and Waste (DWA). Please note that this is a very old recommendation of the United States Water Resources Council in their Bulletin 17B (USWRC, 1982) and was developed in the literature by many other scientists (Ouarda et al. 1998 among many others:

- T. B.M.J. Ouarda, P. F. Rasmussen, B. Bobée et J. Bernier (1998): Use of historical information in hydrologic frequency analysis Vol. 11, Journal of Water Science 41–49
- Ouarda, T., Hamdi, Y., Bobee, B., 2004. A general system for frequency estimation in hydrology (FRESH) with historical data. In: Benito, G., Thorndycraft, V.R. (Eds.), Systematic, Paleoflood and Historical Data for the Improvement of Flood Risk Estimation: Methodological Guidelines. CSIC, Madrid, pp. 55–70.
- Payrastre et al. (2011)…

**Methods for incorporating historical information (lines 145-173)**

In line 145, authors talk about three methods suggested by the German Association for Water Management, Sewage and Waste (DWA) for the incorporation of historical

extremes with systematic observations (DWA, 2012). The two last methods (lines 155-173) were proposed to estimate the distribution functions parameters, are general and have nothing to do with the incorporation of historical information.

**Technical corrections (or clarifications):**

- Journals names are missing in some references:

   Line 515 Haigh et al. (2014b)

   Line 515 Hamdi et al. (2015)

   Line 518 Hastings (1970)

   Line 523 Hinkel et al. (2015)

   Line 527 Jenson et al. (2008)

   And many others…

- Is there a need to detail the Gringorten plotting position formula?
- Authors don't need to detail the DWA methods. Don't need to present the equations since they are not used in the developed method.
- There is a repetition when talking about the benefits of hist information. Many things were repeated in Introduction/section 2.2.
- The choice of the POT threshold must be justified. Why the 98% extreme quantile and why the 1m?
- The choice of the 2 days period in the declustering method must be justified.
- Last paragraph of section 3.2: which type of bootstrap? parametric or non-parametric ? and the choice must be justified.
- Don't need to detail the BIC criteria (don't need to put the equation).